# Connexin 30 Deficiency Ameliorates Disease Progression at the Early Phase in a Mouse Model of Amyotrophic Lateral Sclerosis by Suppressing Glial Inflammation

**DOI:** 10.3390/ijms232416046

**Published:** 2022-12-16

**Authors:** Yu Hashimoto, Ryo Yamasaki, Senri Ko, Eriko Matsuo, Yuko Kobayakawa, Katsuhisa Masaki, Dai Matsuse, Noriko Isobe

**Affiliations:** Department of Neurology, Neurological Institute, Graduate School of Medical Sciences, Kyushu University, Fukuoka 812-8582, Japan

**Keywords:** amyotrophic lateral sclerosis, ALS, SOD1, mSOD1 mice, connexin 30, astrocytes

## Abstract

Connexin 30 (Cx30), which forms gap junctions between astrocytes, regulates cell adhesion and migration, and modulates glutamate transport. Cx30 is upregulated on activated astroglia in central nervous system inflammatory lesions, including spinal cord lesions in mutant superoxide dismutase 1 (mSOD1) transgenic amyotrophic lateral sclerosis (ALS) model mice. Here, we investigated the role of Cx30 in mSOD1 mice. Cx30 was highly expressed in the pre-onset stage in mSOD1 mice. mSOD1 mice with knockout (KO) of the *Cx30* gene (*Cx30*KO-mSOD1 mice) showed delayed disease onset and tended to have an extended survival period (log-rank, *p* = 0.09). At the progressive and end stages of the disease, anterior horn cells were significantly preserved in *Cx30*KO-mSOD1 mice. In lesions of these mice, glial fibrillary acidic protein/C3-positive inflammatory astroglia were decreased. Additionally, the activation of astrocytes in *Cx30*KO-mSOD1 mice was reduced compared with mSOD1 mice by gene expression microarray. Furthermore, expression of connexin 43 at the pre-onset stage was downregulated in *Cx30*KO-mSOD1 mice. These findings suggest that reduced expression of astroglial Cx30 at the early disease stage in ALS model mice protects neurons by attenuating astroglial inflammation.

## 1. Introduction

Amyotrophic lateral sclerosis (ALS) is a severe neurodegenerative disease characterized by the loss of upper and lower motor neurons [1,2,3]. Approximately 10% of individuals with ALS have a family history, and Cu/Zn superoxide dismutase 1 (*SOD1*) was the first gene associated with ALS [4,5]. Mutation of *SOD1* occurs in 12–15% of individuals with familial ALS and 1–2% of individuals with sporadic ALS [6,7].

The neurodegenerative process in mutant SOD1 (mSOD1)-mediated ALS appears to be non-cell autonomous; that is to say, neighboring non-neuronal supporting cells play an essential role in the neuronal dysfunction [8,9,10]. Accumulation of mSOD1 only in neurons did not cause motor neuron impairment [11,12]. Moreover, mice expressing mSOD1 only in astrocytes did not display motor neuron degeneration [13]. These results suggest that accumulation of mSOD1 is essential not just for individual neurons and astrocytes, but also their associations in motor neuron pathology. Chimeric mice composed of mixtures of mSOD1-expressing and normal cells survived longer compared with mSOD1 mice [14]. Moreover, loss of mSOD1 in astrocytes delayed the disease progression of ALS [15,16]. mSOD1-expressing astrocytes secrete toxic factors, such as the transforming growth factor-β1 (TGF-β1) and pro-inflammatory cytokines, which induce motor neuron cell death [17,18,19]. Some of these pro-inflammatory cytokines are released into the extracellular space through hemichannels in astrocytes [20]. Gap junctions play a key role in intercellular communication [21,22], while hemichannels mediate the extracellular transport of various factors [23] and are involved in inflammation [24], cell death [25,26], and calcium homeostasis [27]. 

Gap junctions and hemichannels in astrocytes are formed by connexin 30 (Cx30) and connexin 43 (Cx43) [28]. An investigation employing a tracer in Cx30-deficient mice indicated a role for Cx30 in approximately 20% of interastrocytic couplings in the hippocampus [29]. Inactivation of Cx30 and Cx43 can decrease hippocampal synaptic transmission in astroglial networks. Moreover, these Cxs influence the transport of glutamate and K^+^ buffering [30], and astrocytic Cx has channel-independent functions, such as in cell adhesion [31,32] and intracellular signaling [33]. Mainly, Cx30 modulates hippocampal excitatory and inhibitory synaptic transmission [34,35,36]. Cx30 co-localizes with microtubules and cortical actin filaments [37]. Altogether, these results implicate astrocytic Cx in cell volume regulation and modulation of glutamatergic synaptic activity. 

Accumulating evidence suggests a connection between ALS and connexins [22,38]. Cx43 expression in the spinal cord of SOD1^G93A^ transgenic mice (hereafter referred to as mSOD1 mice) was significantly elevated compared with wild-type (WT) mice [39,40]. Moreover, Cx43 expression is upregulated in postmortem tissue and cerebrospinal fluid of humans with ALS [41]. Cx36, the gap junction protein forming electrical synapses between neurons, is decreased in the lumbar spinal cord of individuals with ALS and mSOD1 mice at the early stage [42,43]. Cx30 immunoreactivity is detected in the gray matter of the spinal cord in mSOD1 mice, but no significant difference between mSOD1 and WT mice was indicated by immunoblotting at the end stage [39,40]. Moreover, no previous report described changes in Cx30 expression at the early stage or its involvement in the pathogenesis of mSOD1 mice.

Reactive astrocytes have two phenotypes: an ‘A1′ proinflammatory phenotype induced by activated neuroinflammatory microglia, and an ‘A2′ neuroprotective phenotype observed after ischemic stroke [44,45,46]. To elucidate the pathogenesis of glial inflammation in neuroinflammatory diseases enacted through Cx30, we previously employed Cx30-deficient mice [47,48]. Cx30 deficiency causes an increase in ramified microglia, enlargement of astrocytic processes in the spinal gray matter, and reduction of Cx43 expression in the spinal white matter. In Cx30-deficient experimental autoimmune encephalomyelitis (EAE) model mice, Cx30 deficiency induced widespread activation of anti-inflammatory phenotype microglia during acute and chronic EAE states. In particular, Cx30 induced earlier and stronger activation of ‘A2′ neuroprotective astrocytes in the acute state. Moreover, a lack of Cx30 decreased Cx43 expression in the chronic state [48].

We hypothesized that the lack of Cx30 led to neuroprotection in ALS model mice from the viewpoint of neuroinflammation. Thus, we aimed to clarify the role of Cx30 using a mouse model of ALS with a knockout (KO) of *Cx30* (*SOD1*^G93A^/*Cx30^−/−^*), hereafter referred to as *Cx30*KO-mSOD1 mice. 

## 2. Results

### 2.1. Cx30 Deficiency Delays Disease Progression in SOD1^G93A^ ALS Mice

We genotyped mice at 4 weeks after birth and conducted experiments on mice aged from 6 weeks to 26 weeks (or death). We classified three stages of the disease course in mSOD1 mice as follows: 8 weeks as the pre-onset stage, 14 weeks as the progressive stage, and 20 weeks as the end stage. The timeline of behavioral experiments is shown in Figure 1. We measured body weights and performed behavioral analyses, including ALS-Therapy Development Institute (TDI) scores, forelimb grip strength, and rotarod tests in mSOD1 (*n* = 14) and *Cx30*KO-mSOD1 (*n* = 14) mice to investigate the behavioral effects of Cx30 deficiency. The time of disease onset was defined as when the body weight peaked retrospectively. The body weight peak in *Cx30*KO-mSOD1 mice was delayed compared with mSOD1 mice (119.0 days vs. 109.6 days, *p* = 0.04; Figure 2A, Appendix A). The timing of the ALS-TDI score to reach 1 (beginning of leg trembling) was delayed in *Cx30*KO-mSOD1 mice compared with mSOD1 mice (*p* < 0.001, 13 weeks to 15 weeks; Figure 2B, 112.0 days vs. 99.4 days, *p* = 0.02; Appendix A). Forelimb grip strength tended to be preserved in *Cx30*KO-mSOD1 mice after 16 weeks, and the difference between the mean forelimb grip strength of *Cx30*KO-mSOD1 mice and mSOD1 mice was largest at 18 weeks, although not statistically significant (*p* = 0.052; Figure 2C). Performance in the rotarod test was significantly reduced in *Cx30*KO-mSOD1 mice compared with mSOD1 mice from 7 weeks to 12 weeks (*p* < 0.001, 7 and 12 weeks; *p* < 0.05, 8 to 11 weeks; Figure 2D). However, there was no difference in the decline in rotarod performance between *Cx30*KO-mSOD1 and mSOD1 animals (118.0 days vs. 105.7 days, *p* = 0.22; Figure 2E, *p* = 0.15; Appendix A). A reduction in performance in the rotarod test was also observed in *Cx30*KO mice compared with WT animals (*p* < 0.05, 9, 10, and 25 weeks; *p* < 0.01, 19, 20, 22, 24, and 26 weeks; *p* < 0.0001, 23 weeks; Appendix A). Forelimb grip strength was similar between *Cx30*KO and WT mice (Appendix A). *Cx30*KO-mSOD1 mice exhibited a relatively longer lifespan than mSOD1 mice (161.2 ± 2.84 days vs. 156.8 ± 1.95 days, *p* = 0.09; Figure 2F). There was a positive correlation between the time of body weight peak and survival time (*p* = 0.03, Appendix A), suggesting that an earlier body weight loss is associated with a more rapid disease progression.

### 2.2. Cx30 Deficiency Decelerates Cell Loss in the Lumbar Spinal Cord of SOD1^G93A^ ALS Mice 

We compared numbers and sizes of lumbar spinal cord neurons at 8 weeks (pre-onset stage), 14 weeks (progressive stage), and 20 weeks (end stage). Immunohistochemistry with an anti-NeuN antibody showed that anterior horn cells gradually decreased over the disease course in mSOD1 and *Cx30*KO-mSOD1 mice (Figure 3A–C). Furthermore, dorsal horn cells were reduced and cells were smaller in mSOD1 and *Cx30*KO-mSOD1 mice at the end stage. The number of anterior horn cells was greater in *Cx30*KO-mSOD1 mice compared with mSOD1 mice at the progressive and end stages (*p* = 0.03 and 0.048, respectively; Figure 3D–F). Dorsal horn cells were more numerous in *Cx30*KO-mSOD1 mice at the pre-onset and end stages (*p* = 0.008 and 0.016, respectively). In addition, anterior and dorsal horn cells were larger in *Cx30*KO-mSOD1 mice than mSOD1 mice at the end stage (*p* < 0.001; Appendix A). Compared with WT mice, cell numbers were significantly decreased in both mSOD1 and *Cx30*KO-mSOD1 mice at the end stage (Figure 3F). Time course analysis showed that Cx30 deficiency had a protective effect on neurons at the end stage (Figure 3G,H; Appendix A).

### 2.3. Cx30 Is Upregulated in mSOD1 Mice at the Pre-Onset Stage

In both mSOD1 and *Cx30*KO-mSOD1 mice, mSOD1 protein was deposited in lumbar anterior horn cells at the pre-onset stage (Appendix A). Based on this result, we hypothesized that the tissue environment in the lumbar spinal cord started to change before disease onset. Therefore, we examined Cx30 expression in the lumbar spinal cord of mSOD1 mice. Cx30 showed marked expression in the spinal gray matter. The spinal cord at the end stage in mSOD1 mice showed loss of Cx30 expression despite high glial fibrillary acidic protein (GFAP) reactivity (Figure 4A). Quantitative Western blot analysis revealed that Cx30 protein expression was similar in 4-week-old mSOD1 and WT mice. Notably, Cx30 protein was significantly upregulated in 8-week-old mSOD1 mice compared with 4-week-old and 20-week-old mSOD1 mice (*p* < 0.0001 and < 0.001, respectively, Figure 4B–D). Western blotting images used for analysis are shown in Appendix A. Cx30 protein levels were similar in 20-week-old mSOD1 and WT mice. These findings suggest that Cx30 plays a role in disease pathogenesis in ALS model mice at the pre-onset stage.

### 2.4. Cx43 Expression in Cx30KO-mSOD1 Mice Is Downregulated at the Pre-Onset Stage

Next, we assessed Cx43 expression in mSOD1 and *Cx30*KO-mSOD1 mice. Cx43 immunoreactivity was more abundant in gray matter than white matter (Figure 4E). Quantitative Western blot analysis showed that Cx43 protein expression was significantly higher in mSOD1 mice compared with *Cx30*KO-mSOD1 mice at the pre-onset stage (*p* = 0.04; Figure 4F,H). In contrast, there was no difference between mSOD1 and *Cx30*KO-mSOD1 mice at 4 weeks, progressive, or end stages (*p* = 0.98, 0.11, and 0.25, respectively; Figure 4G,I,J). Western blotting images used for this analysis are shown in Appendix A. Cx30 deficiency impacted not only Cx30 protein expression but also downregulated Cx43 in the early phase in the spinal cord of ALS model mice.

### 2.5. Cx30 Deficiency Reduces GFAP/C3-Positive Inflammatory Astroglia in SOD1^G93A^ ALS Mice at the End Stage

Next, we performed immunohistochemistry and Western blotting to analyze astrocytic dysregulation in the lumbar spinal cord of *Cx30*KO-mSOD1 mice. GFAP immunoreactivity was mainly detected in the spinal white matter and gradually increased from the anterior horn in spinal gray matter as the disease progressed. Furthermore, expression of S100A10 (a neuroprotective astrocyte marker) and C3 (a proinflammatory astrocyte marker) gradually increased toward the end stage (Figure 5A,B). We measured relative areas of GFAP^+^, C3^+^/GFAP^+^, and S100A10^+^/GFAP^+^ astrocytes in the anterior horns at 8, 14, and 20 weeks (Figure 5C–E). *Cx30*KO-mSOD1 mice had significantly fewer S100A10^+^/GFAP^+^ astrocytes at the pre-onset and progressive stages, and C3^+^/GFAP^+^ astrocytes at the end stage (*p* = 0.005, 0.020, and 0.049, respectively; Figure 5C,D). Moreover, *Cx30*KO-mSOD1 mice had significantly fewer GFAP^+^ astrocytes than mSOD1 mice at the pre-onset and end stages (*p* = 0.036 and 0.043, respectively; Figure 5E). Next, we performed quantitative Western blot analysis of GFAP, C3, and S100A10 in the lumbar spinal cord. S100A10 protein levels were significantly lower in mSOD1 and *Cx30*KO-mSOD1 mice compared with WT mice at the pre-onset stage (*p* = 0.012 and 0.008, respectively; Figure 5F,G), but were significantly higher at the end stage (*p* = 0.013 and 0.002, respectively). C3 protein levels were not significantly different between the three groups at each stage (Figure 5H,I). GFAP protein levels were significantly higher in mSOD1 and *Cx30*KO-mSOD1 mice compared with WT mice at the progressive and end stages (*p* = 0.007, 0.044, 0.003, and 0.008, respectively; Figure 5J,K). Western blotting images used for this analysis are shown in Appendix A. Collectively, these findings indicate that proinflammatory C3^+^ astrocytes were decreased at the end stage in *Cx30*KO-mSOD1 mice compared with mSOD1 mice.

### 2.6. Cx30 Deficiency has No Effect on the Microglial State in SOD1^G93A^ ALS Mice

We performed immunohistochemistry and Western blotting to analyze microglial dysregulation in the lumbar spinal cord of *Cx30*KO-mSOD1 animals. Ionized calcium-binding adapter molecule 1 (Iba1)-positive microglia in mSOD1 and *Cx30*KO-mSOD1 mice had an activated form as opposed to the ramified form observed in WT mice. Numbers of activated microglia in mSOD1 and *Cx30*KO-mSOD1 mice were markedly increased at the end stage (Figure 6A,B). Immunoreactivity of arginase-1 (Arg1) and nitric oxide synthase 2 (NOS2) were visually comparable from the progressive to end stages among all genotypes (Figure 6C,D). For quantitative analysis of microglia, we performed Western blotting for Iba1, Arg1, and NOS2 (Figure 6E–J). At the end stage, Iba1 protein levels were significantly higher in mSOD1 and *Cx30*KO-mSOD1 mice compared with WT mice (*p* = 0.020 and 0.013, respectively; Figure 6E,F). Arg1 protein levels were comparable among mSOD1, *Cx30*KO-mSOD1, and WT mice (Figure 6G,H). However, NOS2 protein levels were significantly lower in mSOD1 and *Cx30*KO-mSOD1 mice at the end stage (*p* = 0.01 and 0.04, Figure 6I,J). Western blot images used for analysis are shown in Appendix A. In summary, the microglial status was similar between *Cx30*KO-mSOD1 mice and mSOD1 mice.

### 2.7. Cx30 Deficiency Reduces Astrocyte Activation in SOD1^G93A^ ALS Mice as Assessed by Gene Expression Microarray

Next, we performed an RNA array assay to further evaluate the dysregulation of astrocytic gene expression in the spinal cords of *Cx30*KO-mSOD1 mice at 8 weeks. We detected pro- and anti-inflammatory gene sets [49], as well as pan-reactive (PAN), A1-specific, and A2-specific genes [45]. We then performed a heatmap/cluster analysis. In line with the results of immunohistochemical analysis, inflammatory astrocyte-related genes were downregulated in *Cx30*KO-mSOD1 mice (Figure 7A,B). In comparison, anti-inflammatory genes were not altered in a consistent manner—some genes were upregulated, while others were downregulated (Figure 7A,B). These results indicate that *Cx30*KO-mSOD1 mice had reduced astrocyte activation compared with mSOD1 mice.

## 3. Discussion

The main findings of the present study are as follows: (1) Cx30 deficiency delayed disease onset, relatively prolonged survival, and significantly attenuated the loss of lumbar neurons to mSOD1 mice. (2) Cx30 expression was increased at the pre-onset stage in mSOD1 mice. (3) Cx30 deficiency suppressed Cx43 expression in the lumbar spinal cord of ALS model mice at the pre-onset stage. (4) Cx30 deficiency diminished the activation of astrocytes in ALS model mice.

The relationship between Cx30 and neurodegenerative disease is not well documented. In the β-amyloid precursor protein/presellin1 mouse model of Alzheimer’s disease, Cx30 immunoreactivity was increased in reactive astrocytes associated with amyloid plaques [50]. In the brains of human patients with Alzheimer’s disease, Cx30 immunoreactivity was increased in reactive astrocytes at amyloid plaques [51]. In addition, Cx30 expression was increased in perivascular astrocytic endfeet of two Parkinson’s disease model animals, 6-hydroxydopamine-exposed rats and 1-methyl-4-phenyl-1,2,3,6-tetrahydropyridine (MPTP)-exposed monkeys [52]. Cx30-deficient mice had more accelerated dopaminergic neuron loss than WT mice following exposure to MPTP, and expression of PAN and A2 astrocyte genes was reduced in the striatum of Cx30-deficient mice compared with WT following MPTP exposure [53]. In ALS model mice, Cx30 protein and mRNA expression levels were similar to WT mice at the pre-symptomatic, disease-progressive, and end stages [39]. However, immunohistochemistry of the lumbar spinal cord in end-stage mSOD1 mice displayed a patchy loss of Cx30 expression [40]. This study differs from our previous study because here we focused on the early phase and found that Cx30 was significantly upregulated in mSOD1 animals at an early pre-symptomatic stage. This upregulation of Cx30 might be caused by mSOD1 protein accumulation and reactive astrocytosis in anterior horn cells at the disease pre-onset stage.

Cx30 and Cx43, two glial-specific connexins, have different features. Cx30 is expressed on astrocytes, while Cx43 is expressed on astrocytes and microglia [38]. Cx30 is expressed in the cerebral white matter at very low levels, and is mainly observed in Bergmann glia and astrocytes in the cerebellar cortex and spinal gray matter [54]. Cx43 is highly expressed during embryonic and postnatal development, whereas Cx30 is expressed 3 weeks after birth [55,56,57]. The half-life of Cx30 is longer than other connexins [58]. In astrocyte/neuron cocultures, Cx30 is induced in astrocytes proximal to the neuronal soma [57]. In this study, Cx30 expression declined as the disease progressed, regardless of the increase in GFAP^+^/C3^+^ astrocytes. Moreover, less co-localization of Cx30 with GFAP^+^ astrocytes was observed compared with Cx43. These findings suggest that Cx30 is downregulated, while Cx43 is upregulated in reactive astrocytes.

This study showed that Cx43 was upregulated in mSOD1 mice at the pre-onset stage, though this upregulation was suppressed in *Cx30*KO-mSOD1 mice compared with mSOD1 mice. In our previous study of EAE, *Cx30*KO mice had less Cx43 immunoreactivity in the spinal white matter during the chronic stage [48]. Moreover, hippocampal immunoblotting of *Cx30*KO mice revealed no upregulation of Cx43 expression [29]. Thus, Cx30 deficiency can influence and downregulate Cx43 in the spinal cord. Several previous reports have described the suppression of Cx43 as neuroprotective. For example, spinal cord injury model rats treated with an Cx43-antisense oligodeoxynucleotide had clinical locomotor improvement, less swelling of the spinal cord, less upregulation of GFAP, and reduced microglial activation [59]. In addition, experiments using Cx43/Cx30 double-KO mice showed that neuropathic pain due to spinal cord injury was more prevented in Cx43/Cx30 double-KO mice than *Cx30*KO mice [60]. In a motor neuron and SOD1^G93A^ astrocyte co-culture system, motor neurons were preserved by inhibiting Cx43 using the Cx43 blocker GAP26 and Cx43 hemichannel blocker GAP19 [40]. Overexpression of Cx43 induces neuroglial inflammation through the extracellular release of glutamate [20], ATP [61], D-serine [62], prostaglandin E2 (PGE2) [63], and nicotinamide adenine dinucleotide [64]. Cx43 deficiency results in reduced hemichannel-mediated activity and calcium levels, which lead to neuroprotection [40,41,65]. Therefore, the suppression of Cx43 induced by Cx30 deficiency may contribute to the neuroprotective effect observed in the present study.

In *Cx30*KO-mSOD1 mice, gene expression levels of chemokines such as C-C motif chemokine ligand (CCL) 2, CCL5, C-X-C motif chemokine ligand (CXCL) 10, and SerpinA3n, a reactive astrocyte marker, were lower than in mSOD1 mice. Additionally, based on immunohistochemistry, reactive astrocytosis was reduced at the end stage. Reactive astrocytes reportedly express many detrimental actions in ALS [66,67]. TGF-β1 secreted by reactive astrocytes suppressed motor neuron branching and promoted protein aggregation in motor neurons by inducing autophagy defects [19]. Reactive astrocytes exposed to cerebrospinal fluid from patients with ALS (ALS-CSF) released interleukin (IL)-6, tumor necrosis factor α, cyclo-oxygenase 2, and PGE2 though downregulated IL-10, vascular endothelial growth factor, and glial cell line-derived neurotrophic factor. Moreover, astroglia exposed to ALS-CSF released high levels of glutamate, reactive oxygen species, and nitric oxide [18]. Reactive astrocytes suppressed the activity of D-amino acid oxidase, contributing to an increase in D-serine in mSOD1 mice [68]. Motor neurons co-cultured with ALS astrocytes displayed a loss of major histocompatibility complex class I (MHCI), which was caused by endoplasmic reticulum stress related to susceptibility to reactive astrocyte-induced toxicity [69]. The mechanism by which astrocytes become reactive includes the calcineurin/nuclear factor of activated T-cells (CN/NFAT), Janus kinase/signal transducer and activator of transcription-3 (JAK/STAT3), nuclear factor of kappa light polypeptide gene enhancer in B cells (NF-κB), and mitogen-activated protein kinase (MAPK) pathways. Of these pathways, only the CN/NFAT pathway is activated by glutamate [70]. A lack of Cx30 and Cx43 may induce a reduction of glutamate transport, resulting in a decrease of reactive astrocytes.

As described above, our results indicate that Cx30 deficiency has a neuroprotective role in ALS model mice. However, the survival of *Cx30*KO-mSOD1 mice was not significantly extended compared with mSOD1 mice, possibly because the toxicity of mSOD1 exceeds the effects of Cx30 deficiency. In addition, there are several limitations to this study. First, in the behavioral study, reductions in grip strength and rotarod test performance were similar among *Cx30*KO-mSOD1 and mSOD1 mice. In terms of motor dysfunction in mSOD1 mice, limb palsy starts from the hindlimb and extends to the forelimb [71]. Our results show that the difference in forelimb grip strength between *Cx30*KO-mSOD1 and mSOD1 mice was insignificant. However, the result of ALS-TDI scores, an indicator of hindlimb signs, showed that *Cx30*KO-mSOD1 mice significantly delayed hindlimb suffering compared with mSOD1 mice. These results indicate that Cx30 deficiency affected the early phase of suffering in the hindlimb but had only a low impact on the late phase extending to the forelimb. The performances of *Cx30*KO-mSOD1 mice in the rotarod test were significantly lower in the initial stage compared with mSOD1 mice. However, low scores in the rotarod test might not be caused by motor dysfunction because the grip strength of *Cx30*KO-mSOD1 mice was similar to mSOD1 mice, and *Cx30*KO-mSOD1 mice showed no gait disturbance or ataxia. A previous study showed that *Cx30*KO mice exhibit reduced exploratory activity (rearing) but not locomotion, in the open field test. Moreover, *Cx30*KO mice exhibited anxiogenic behaviors, including higher open-field center avoidance and corner preference [72]. These behaviors may have contributed to falling off the rotating rod earlier, resulting in low test scores. Therefore, the lack of a difference between *Cx30*KO-mSOD1 and mSOD1 mice at the peak of the rotarod test may be related to the phenotype of *Cx30*KO mice. In addition, the fact that rotarod test scores of *Cx30*KO-mSOD1 mice were more significantly varied than scores of mSOD1 mice also may be related **(**Appendix A). Second, in our assessment of astrocytes, some differences observed in immunohistochemistry were not detected by Western blot. This difference may be caused by only analyzing the anterior horn by immunohistochemistry, whereas the whole lumbar spinal cord was used for Western blots.

Taken together, Cx30 was upregulated at the earlier pre-symptomatic stage in the lumbar spinal cord of mSOD1 mice. A lack of Cx30 induced a decrease in Cx43 expression and suppressed astroglial inflammation at the same stage, which delayed the onset of hindlimb signs and body weight loss, and reduced neuronal cell loss in the lumbar spinal cord. Surprisingly, Cx30 deficiency did not affect the state of microglia in mSOD1 mice, suggesting that the neuroprotective role of Cx30 deficiency in ALS model mice results from a different and unique pathology from our previous report of Cx30-deficient EAE model mice [48].

## 4. Materials and Methods

### 4.1. Ethical Statement

All efforts were made to minimize the number and suffering of mice based on guidelines for the proper conduct of animal experiments published by the Science Council of Japan and the ARRIVE guidelines 2.0 (Animal Research: Reporting of In Vivo Experiments) for animal research [73,74]. The Animal Care and Use Committee of Kyushu University granted ethical approval for the study on 20 December 2019 (#A19-367) and permitted an extension of the experimental period until 31 March 2023 (#A21–111).

### 4.2. Mice and Genotyping

Transgenic mice with the human *SOD1*^G93A^ gene were purchased from the Jackson Laboratory (Bar Harbor, ME, USA) and backcrossed to C57BL/6J mice for more than 20 generations. *Cx30*KO mice were purchased from the European Mouse Mutant Archive and backcrossed to C57BL/6J mice. 

Heterozygous *SOD1*^G93A^ male mice (*n* = 2) and *Cx30*KO female mice (*n* = 4) were crossed to obtain *SOD1*^G93A^/*Cx30*^−/WT^ mice (*n* = 20) and *Cx30*^-/WT^ mice (*n* = 20). Next, *SOD1*^G93A^/*Cx30*^−/WT^ male mice (*n* = 10) and *Cx30*^−/WT^ female mice (*n* = 20) were crossed to obtain *SOD1*^G93A^/*Cx30*^−/−^ mice (*n* = 50) and *SOD1*^G93A^/*Cx30*^WT/WT^ mice (*n* = 50). *Cx30^WT/WT^* mice and *Cx30*^−/−^ mice were used as standard controls for comparison with *SOD1^G93A^* mice (*n* = 50 for each genotype). For the behavioral study, male mice with *SOD1^G93A^/Cx30^−/−^* (*n* = 14), *SOD1^G93A^/Cx30^WT/WT^* (*n* = 14), *Cx30^−/−^* (*n* = 10), and *Cx30^WT/WT^* (*n* = 10) genotypes were used. Moreover, male and female mice of all genotypes were used for immunohistochemistry (*n* = 15) and Western blotting (*n* = 20). Littermates not used for experiments and mice for breeding were euthanized by anesthesia with isoflurane at 4 and 20 weeks, respectively. The total number of mice used in experiments was 486. All animals were maintained in an air-conditioned, specific-pathogen-free room with a time-controlled lighting system in the Center of Biomedical Research, Research Center for Human Disease Modeling, Graduate School of Medical Sciences, Kyushu University. 

Transgenic mice for the human *SOD1*^G93A^ gene and *Cx30*KO mice were genotyped by PCR of DNA obtained from ear punches. Primer pairs for detecting mutant SOD1 were SOD1-Ex4 (F): 5ʹ-CAT CAG CCC TAA TCC ATC TGA-3ʹ and SOD1-Ex4 (R): 5ʹ-CGC GAC TAA CAA TCA AAG TGA-3ʹ. The mSOD1 DNA product was 236 bp in length. Primer pairs for detecting *Cx30*KO were *Cx30*KO-1: 5ʹ-GGT ACC TTC TAC TAA TTA GCT TGG-5ʹ; *Cx30*KO-2: 5ʹ-AGG TGG TAC CCA TTG TAG AGG AAG-3ʹ; and *Cx30*KO-3: 5ʹ-AGC GAG TAA CAA CCC GTC GGA TTC-3ʹ. The *Cx30*KO DNA product was 460 bp and the WT product was 544 bp.

### 4.3. Behavioral Study

Experiments for analyzing the phenotype of *Cx30*KO-mSOD1 mice included measurements of body weight and grip strength, the rotarod test, and ALS-TDI neurological scoring. We analyzed 14 *Cx30*KO-mSOD1 mice, 14 mSOD1 mice, 10 WT mice, and 10 *Cx30*KO mice. Experiments were conducted on mice aged 6–26 weeks (or until death) at the same weekly time. The definition of death for mSOD1 and *Cx30*KO-mSOD1 mice was the time at which food intake was reduced due to astasia. Timing of onset was defined as the time when body weight peaked, retrospectively [75].

#### 4.3.1. Grip Strength

Forelimb grip strength was assessed with a grip strength meter for mice (MK-380M, Muromachi-Kikai, Tokyo, Japan). Mice were held by the tail and placed on the net of the grip strength machine, which pulled the tail after their forelimb gripped the machine’s net. Mice were gently pulled away until they released the net. Five trials were performed and the best score (in grams) among the five was recorded [76,77].

#### 4.3.2. Rotarod Test

Rotarod test assessed motor coordination, hindlimb strength, and balance. First, mice were trained on the rotarod apparatus (47650 Mouse Rotarod NG, Ugo Basile, Gemonio, Italy) at a fixed speed of 5 rpm. This training was only performed on the first day. For the trial, the apparatus was set at an acceleration speed of 5 to 30 rpm over 300 s referring to the previous accelerating rod protocol [76,77,78]. Mice were placed onto the drum of the apparatus and the time when they fell off the drum was recorded. The trial lasted for a maximum of 300 s. Three trials were performed and the longest time of the three trials was accepted.

#### 4.3.3. ALS-TDI Scoring

To assess ALS-TDI score, traits were registered of the mice while they were suspended by their tails and walking. They were held over the wire top of their cage and their hindlimbs observed. The suspension was repeated 3 times and the most consistent result was recorded. The ALS-TDI neurological score was assessed as follows: 0, normal spray and normal gait; or 1, hindlimb collapsed toward the lateral midline, trembled, or retracted. The gait was normal or slightly slow [79,80,81,82].

### 4.4. Immunohistochemical and Immunofluorescence Analyses

Mice were deeply anesthetized with isoflurane and perfused with phosphate-buffered saline (PBS) and 4% paraformaldehyde (PFA). The lumbar cord was dissected and immersed in 4% PFA at 4 °C overnight, and then sequentially immersed in 20% and 30% sucrose in PBS at 4 °C overnight. Samples were embedded in Tissue-Tek O.C.T. compound (4583, Sakura Finetek, Tokyo, Japan) and stored at −80 °C. Frozen samples were cut into 20-µm and 40-µm slices on a cryostat. Sections were collected in PBS and stored at 4 °C. Cross sections (20 µm) were used for immunohistochemical staining. Endogenous peroxidase activity was quenched with 3% H_2_O_2_ in methanol and PBS (1:1) for 30 min at room temperature. Sections were blocked with Block Ace (KAC, Hyogo, Japan) for 30 min at room temperature and then incubated with primary antibodies at 4 °C overnight. The enhanced indirect immunoperoxidase method was employed using an Envision kit (Agilent Technologies, Santa Clara, CA, USA). Immunoreactivity was detected using 3,3ʹ-diaminobenzidine-tetrahydrochloride (Vector Laboratories, Newark, CA, USA). Axial sections (40 µm) were used for immunofluorescent staining. Sections were incubated with primary antibodies at 4 °C overnight after blocking with Block Ace for 30 min. Next, sections were incubated with secondary antibodies conjugated to Alexa Fluor 488 or 594 (1:300; Thermo Fisher Scientific, Waltham, MA, USA) and mounted with VECTASHIELD HardSet (H-1400, Vector Laboratories). Tissues were observed with a laser-scanning confocal microscopy system (Nikon A1, Nikon, Tokyo, Japan). Primary antibodies are listed in Appendix A.

### 4.5. Quantification of Immunohistochemistry and Immunofluorescence Results

Immunohistochemistry and immunofluorescence results were quantified using ImageJ bundled with JAVA 1.8.0_172 (Mac OS X version of NIH Image; downloaded from https://imagej.nih.gov/ij/download.html (accessed on 1 September 2019)) using three-to-five lumbar spinal cord sections from three-to-six mice in each group. For the quantification of the cells in the lumbar spinal cord, immunohistochemical staining with anti-NeuN antibody was performed. Each section was divided into halves with a vertical line through the central canal. The anterior horn was defined as a ventral part surrounded by a vertical and horizontal line through the central canal, and the dorsal horn was defined as its dorsal part. The size of the surrounding region and the number and average size of cells were calculated with ImageJ, and the cell density (/mm^2^) was manually calculated. For quantification of GFAP, S100A10, and C3 signals, the fluorescence images from one side of the anterior horn in the lumbar spinal cord were analyzed. Multiple immunofluorescence images of the lumbar spinal cord were cut from the background and divided into red images (GFAP) and red images that overlapped with green images (S100A10 and C3). Images were set to the same threshold to measure proportions of immunostained areas.

### 4.6. Western Blotting

Mouse lumbar spinal cords were homogenized on ice in 5 µL/mg lysis buffer containing 0.1% sodium dodecyl sulfate and a protease inhibitor cocktail (Nacalai Tesque, Kyoto, Japan). Lysates were incubated on ice for 30 min and then centrifuged at 4 °C for 10 min at 14,000× *g*. The supernatants were collected, and protein concentrations were determined with a BCA Protein Assay Kit (Thermo Fisher Scientific). Protein samples diluted to 2 µg/µL were heated at 95 °C for 5 min. Samples were separated using 4–15% mini-Protean TGX precast protein gels (Bio-Rad, Hercules, CA, USA) and transferred onto polyvinyl difluoride membranes. The membranes were blocked with 5% skim milk for 1 h at room temperature, followed by primary antibodies diluted in 3% skim milk at 4 °C overnight. The membranes were thereafter incubated for 1 h at room temperature with horseradish peroxidase-conjugated secondary antibodies diluted in 1% skim milk for Cx30, and NOS2 or Tris-buffered saline with Tween 20 for the others. The membrane bands were detected with ECL Prime (Cytiva, Tokyo, Japan). The intensities of acquired bands were measured using Image Lab 6.0 (Bio-Rad) and standardized to β-actin. Primary antibodies are listed in Appendix A.

### 4.7. Gene Expression Microarray

Following the manufacturer’s protocol, we isolated total RNA from dissected spinal cord samples using an RNeasy Mini Kit (Qiagen, Hilden, Germany) and quantified it with an ND-1000 spectrophotometer (NanoDrop Technologies, Wilmington, DE, USA). RNA quality was assessed with a 4150 TapeStation (Agilent Technologies, Santa Clara, CA, USA). Total RNA (50 ng) was then amplified, labeled with an Agilent Low-Input QuickAmp Labeling Kit (Agilent Technologies), and hybridized to SurePrint G3 Mouse GE microarray 8 × 60K ver. 2.0 (Agilent Technologies) according to the manufacturer’s protocols. All hybridized microarrays were scanned with a Microarray Scanner (Agilent Technologies). We calculated relative hybridization intensities and background hybridization values using Feature Extraction software (9.5.1.1) (Agilent Technologies). The gene array results were uploaded to the Gene Expression Omnibus repository (Accession number: GSE213844) at the National Center for Biotechnology Information homepage (https://www.ncbi.nlm.nih.gov/geo/query/acc.cgi?acc=GSE213844 (accessed on 24 September 2022)).

### 4.8. Data Analysis and Filter Criteria

According to the procedures recommended by Agilent, the raw signal intensities and Flags for each probe were calculated from hybridization intensities (gProcessedSignal) and spot information, including glsSaturated. The Flag criteria in GeneSpring Software were as follows: Absent (A) as “Feature is not positive and significant” and “Feature is not above background”; Marginal (M) as “Feature is not Uniform”, “Feature is Saturated”, and “Feature is a population outlier”; Present (P). The raw signal intensities of two samples were log_2_-transformed and normalized with a quantile algorithm using the preprocessCore library package in Bioconductor software [83,84]. Next, we selected probes with the P flag for at least one sample. We calculated Z-scores and ratios (non-log scaled fold-change) from each probe’s normalized signal intensities for comparison between control and experimental samples to identify upregulated and downregulated genes [85]. We used the following criteria: upregulated genes had Z-scores ≥ 2.0 and ratios ≥ 1.5-fold, and downregulated genes had Z-scores ≤ −2.0 and ratios ≤ 0.66. After estimating the statistical significance of the enrichment score (ES), we calculated the false discovery rate (FDR). The ES was considered significant when the normalized *P*-value was < 0.05 and the FDR was < 0.25. Heat maps were generated using the pheatmap package (https://cran.rstudio.com/bin/windows/contrib/3.5/pheatmap_1.0.12.zip (accessed on 20 August 2022)) or heatmap.2 (https://cran.r-project.org/web/packages/gplots/ (accessed on 20 August 2022)) in R software. Briefly, after log_2_-transformation of the original mRNA signal value, the distance from each gene median value (control) was calculated. The log_2_-transformed distance from each gene median value was represented in a color gradient on the heat map. If the log_2_-transformed distance was more significant than 2, the color was changed to that for 2. Similarly, if the log_2_- transformed distance was less than −2, the color was changed to that for −2. Subsequently, the numbers −2 to 2 were placed beside the color bar for the heat map to indicate the log_2_-transformed distances, reflecting fold differences in gene expression from less than 0.25 to more than 4, respectively.

### 4.9. Statistical Analysis

Data are expressed as the mean ± standard error of the mean (SEM). The Gehan − Breslow − Wilcoxon test was used to statistically compare the timing of onset and ALS-TDI score = 1. Multiple t-tests were used to statistically compare the ALS-TDI score, grip strength data, and rotarod test data. The log-rank test was used to statistically compare declines in the rotarod test and survival time. Simple linear regression was used to analyze the relationship between survival time and the timing of body weight peak. An unpaired *t*-test with Welch’s correction was used to analyze the lumbar horn cell numbers and size at 8 and 14 weeks. Cell numbers and size at 20 weeks were analyzed by Welch’s ANOVA plus Dunnett’s T3 multiple comparison test. Comparisons of lumbar spinal cord cells between mSOD1 and *Cx30*KO-mSOD1 mice at each stage were performed by two-way repeated-measures ANOVA with Sidak’s multiple comparison test. One-way ANOVA was used to assess Cx30 and Cx43 expression levels with Tukey’s multiple comparison test. An unpaired *t*-test with Welch’s correction was used to assess the area fractions of GFAP^+^, S100A10^+^/GFAP^+^, and C3^+^/GFAP^+^ cells. One-way ANOVA with Tukey’s multiple comparison test was used to evaluate levels of GFAP, S100A10, C3, Iba1, Arg1, and NOS2. All statistical analyses were performed using Graph Pad Prism 9.0 software (GraphPad Software, San Diego, CA, USA).

## 5. Conclusions

Cx30 accelerated astroglial inflammation by inducing the activation of astrocytes toward a neuroinflammatory pathway in ALS model mice. Our findings suggest that targeting the dysregulation of gap junctions/hemichannels is a novel therapeutic strategy for ALS. Here, we mainly investigated the effect of Cx30 deficiency on glial inflammation in mSOD1 mice. Future studies should examine the relationship between the Cx30 deficit and increase in gap junction hemichannels.

## Figures and Tables

**Figure 1 ijms-23-16046-f001:**
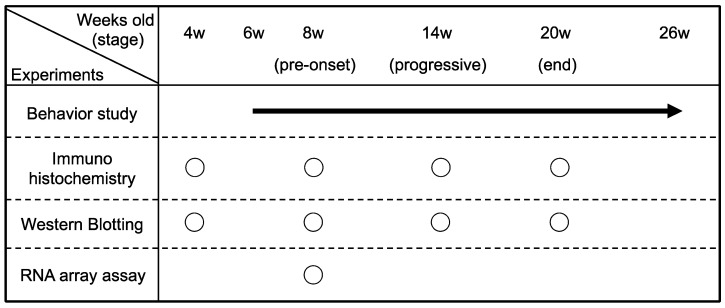
The timeline of experiments. We defined 8, 14, and 20 weeks old as the pre-onset, progressive, and end stages, respectively. The behavioral study was performed from 6 weeks until death. Immunohistochemistry and Western blotting were analyzed at 4, 8, 14, and 20 weeks. An RNA array was performed at 8 weeks.

**Figure 2 ijms-23-16046-f002:**
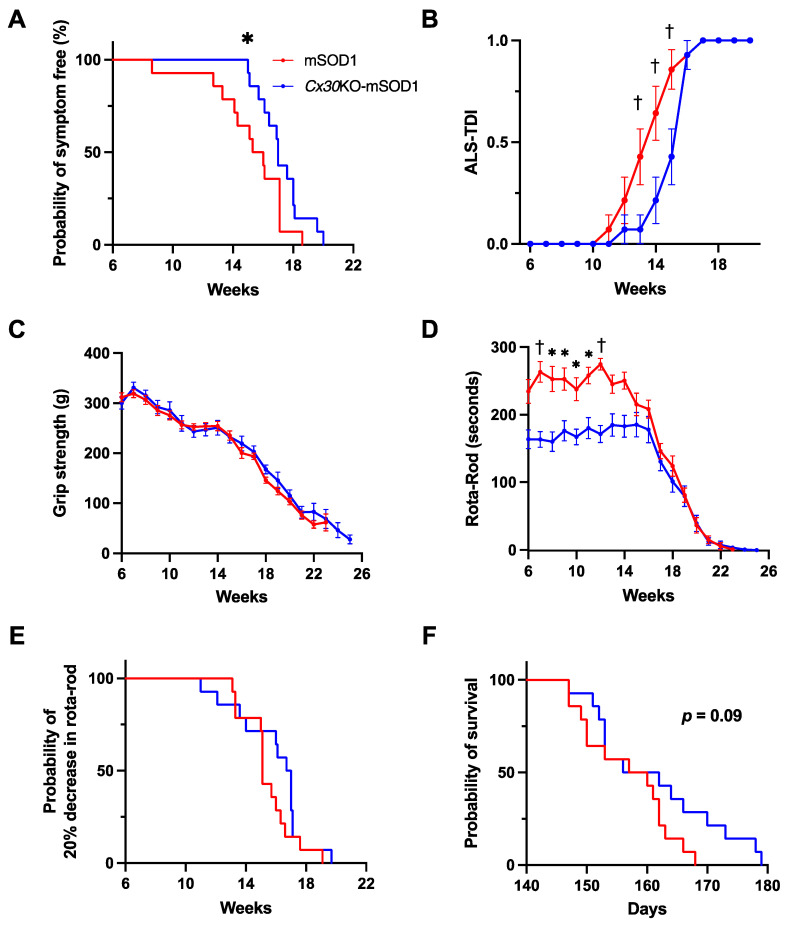
Phenotypic features of *Cx30*KO-mSOD1 mice and mSOD1 mice. (**A**) Symptom-free survival curve, (**B**) ALS-TDI score, (**C**) grip strength, (**D**) rotarod test, (**E**) curve of the decline in rotarod test performance, and (**F**) survival curve (*Cx30*KO-mSOD1 mice and mSOD1 mice, *n* = 14). Red and blue lines indicate mSOD1 and *Cx30*KO-mSOD1 mice, respectively. Means and SEM are shown. The Gehan−Breslow−Wilcoxon test was used to statistically compare the timing of onset as the timing of body weight peak. Multiple t-tests were used to statistically compare ALS-TDI scores, grip strength data, and rotarod test data. The log-rank test was used to compare declines in rotarod test performance and survival time (*p* = 0.22 and 0.09, respectively). * *p* < 0.05 and † *p* < 0.001.

**Figure 3 ijms-23-16046-f003:**
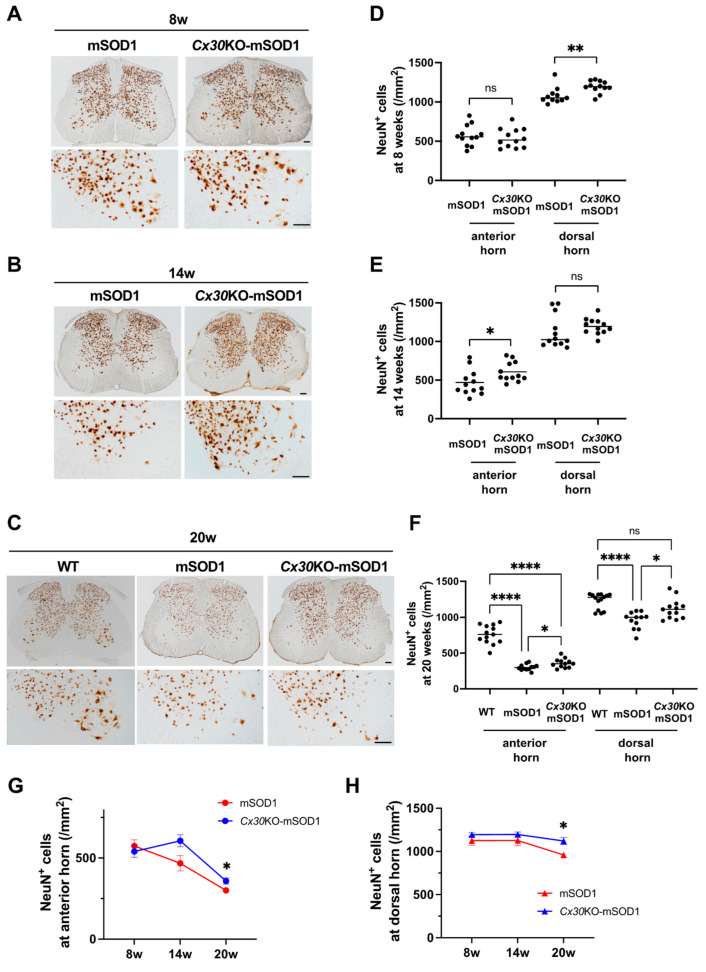
Quantification of lumbar spinal neurons in *Cx30*KO-mSOD1 and mSOD1 mice. (**A**–**C**) NeuN immunostaining of lumbar neurons in *Cx30*KO-mSOD1 and mSOD1 mice at 8 weeks (A), 14 weeks (**B**), and 20 weeks (**C**). NeuN immunostaining at 20 weeks includes WT mice (**C**). (**D**–**F**) Numbers of neurons in the anterior and dorsal horn at each week of age. *n* = 12 (four slices from every three mice). (**G**,**H**) Time course of anterior and dorsal horn cell numbers in each group. Red and blue lines show mSOD1 and *Cx30*KO-mSOD1 mice, respectively. Scale bars: 100 µm. Statistical differences were assessed using an unpaired *t*-test with Welch’s correction for anterior and dorsal horn cell numbers at 8 and 14 weeks. At 20 weeks, Welch’s ANOVA plus Dunnett’s T3 multiple comparison tests were used to analyze anterior and dorsal horn cell numbers. Time course of NeuN^+^ cell numbers was evaluated with two-way ANOVA plus Sidak’s multiple comparison test. ns = not significant. * *p* < 0.05, ** *p* < 0.01, and **** *p* < 0.0001.

**Figure 4 ijms-23-16046-f004:**
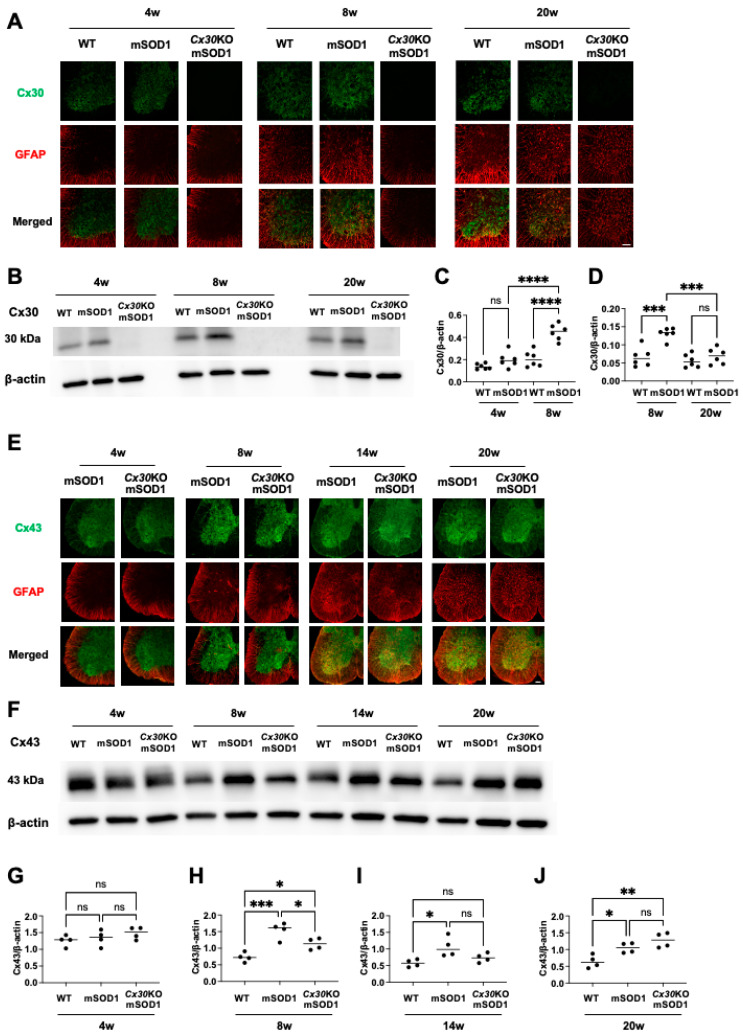
Immunohistochemical and Western blot analysis of connexin dysregulation in the lumbar spinal cord. (**A**) Representative image of immunostaining for Cx30 and GFAP in the lumbar spinal cord of WT, mSOD1, and *Cx30*KO-mSOD1 mice at 4, 8, and 20 weeks. (**B**) Representative Western blot images of Cx30 and β-actin in WT, mSOD1, and *Cx30*KO-mSOD1 mice at 4, 8, and 20 weeks. (**C**,**D**) Western blot analysis of Cx30 protein levels normalized to β-actin levels. *n* = 6. (**E**) Representative image of immunostaining for Cx43 and GFAP in the lumbar spinal cords of mSOD1 and *Cx30*KO-mSOD1 mice at 4, 8, 14, and 20 weeks. (**F**) Representative Western blot images of Cx43 and β-actin in WT, mSOD1, and *Cx30*KO-mSOD1 mice at 4, 8, 14, and 20 weeks. (**G**–**J**) Quantification of Western blot data of Cx43 protein levels normalized to β-actin in each group. *n* = 4. Scale bars: 100 µm. Statistical differences were determined using one-way ANOVA plus Tukey’s multiple comparison test. ns = not significant. * *p* < 0.05, ** *p* < 0.01, *** *p* < 0.001, and **** *p* < 0.0001.

**Figure 5 ijms-23-16046-f005:**
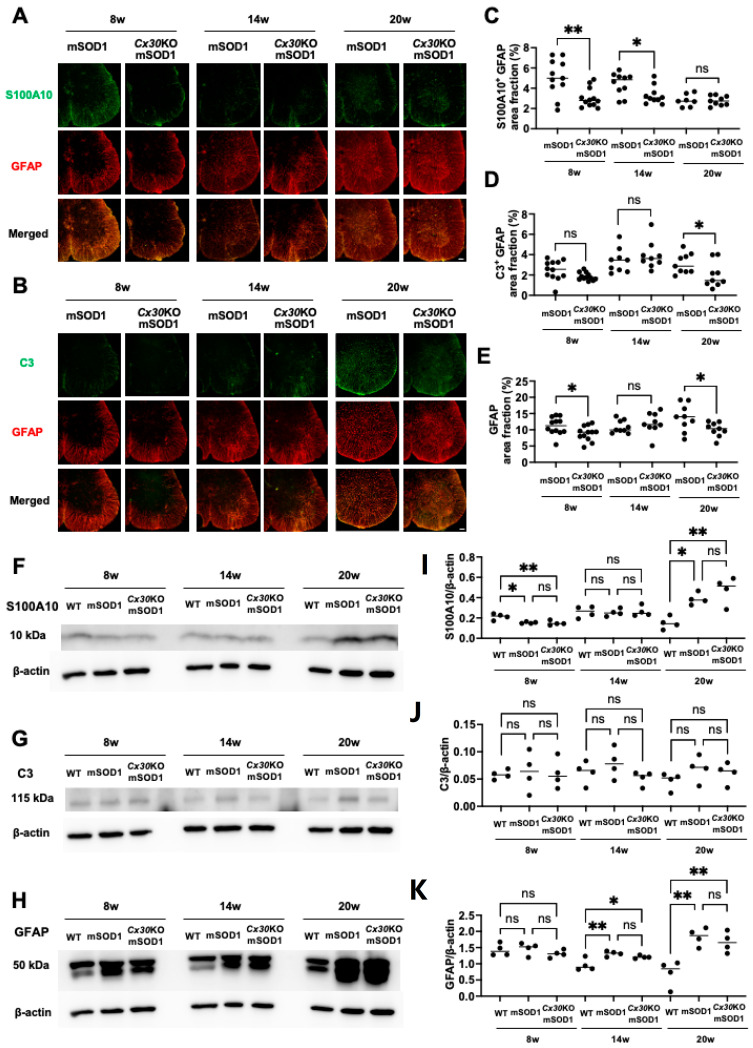
Immunohistochemical and Western blot analysis of astrocytic dysregulation in the lumbar spinal cords of ALS model mice at different weeks of age. (**A**,**B**) Representative images of mice lumbar spinal cords stained with the indicated marker at 8, 14, and 20 weeks. (**C**–**E**) Quantification of relative areas of S100A10^+^/GFAP^+^ cells, C3^+^/GFAP^+^ cells, and GFAP^+^ cells at 8, 14, and 20 weeks. 8 weeks, *n* = 12 (four slices from each of three mice); 14 and 20 weeks, *n* = 9 (three slices from each of three mice). Statistical differences between mice of the same age were assessed using an unpaired *t*-test with Welch’s correction for relative areas. (**F**–**K**) Representative images and Western blot data for S100A10, C3, and GFAP protein levels normalized to β-actin levels at 8, 14, and 20 weeks. *n* = 4. Scale bars: 100 µm. Statistical differences between mice of the same age were assessed using one-way ANOVA plus Tukey’s multiple comparison test for Western blot. ns = not significant. * *p* < 0.05 and ** *p* < 0.01.

**Figure 6 ijms-23-16046-f006:**
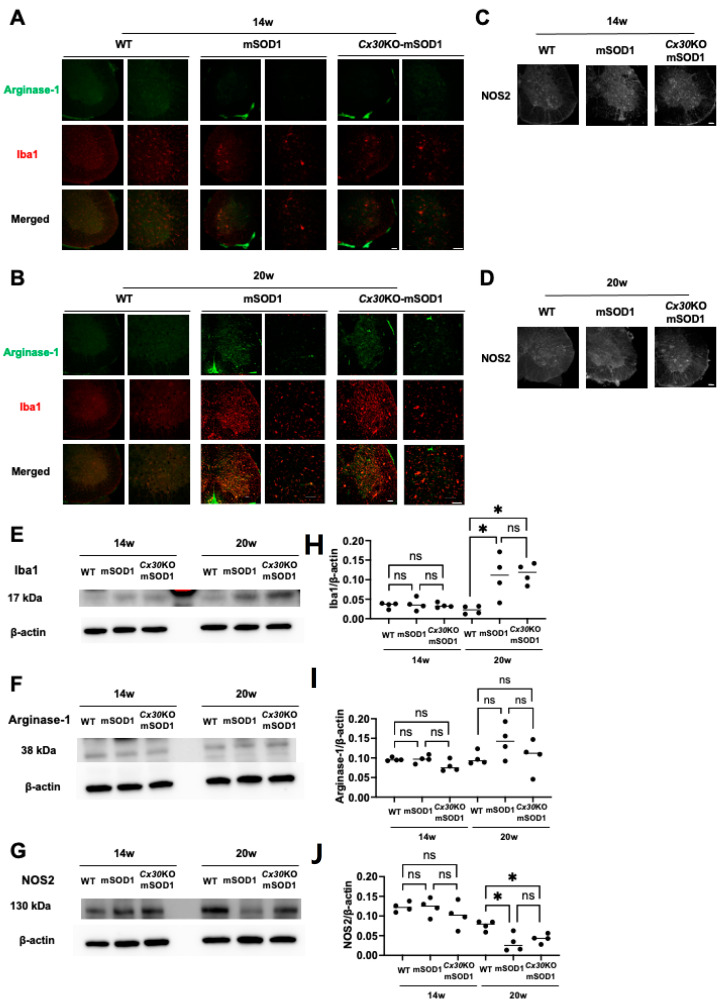
Immunohistochemical and Western blot analysis of microglial activation in the lumbar spinal cords of ALS model mice at different weeks of age. (**A**,**B**) Representative images of immunostaining for Iba1 and Arg1 in the lumbar spinal cords of WT, mSOD1, and *Cx30*KO-mSOD1 mice at 14 and 20 weeks. (**C**,**D**) Representative image of immunostaining for NOS2 in the lumbar spinal cords of WT, mSOD1, and *Cx30*KO-mSOD1 mice at 14 and 20 weeks. (**E**–**J**) Representative Western blot images and analysis of Iba1, Arg1, and NOS2 protein levels normalized to β-actin at 14 and 20 weeks. *n* = 4. Scale bars: 100 µm. Statistical differences were evaluated using one-way ANOVA plus Tukey’s multiple comparison test. ns = not significant. * *p* < 0.05.

**Figure 7 ijms-23-16046-f007:**
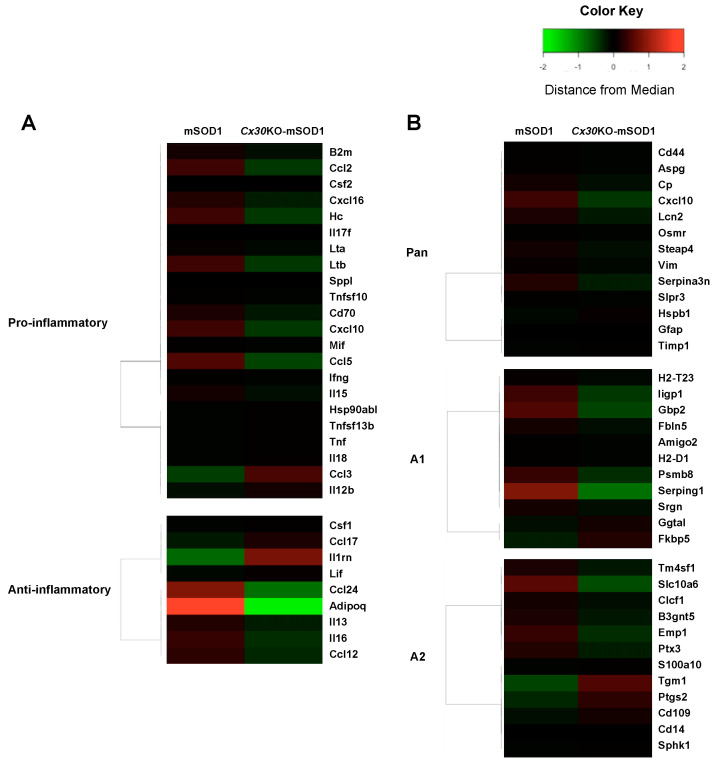
Microarray analysis of spinal cords from mSOD1 and *Cx30*KO-mSOD1 mice. (**A**) Cluster analysis of gene expression arrays of pro-inflammatory and anti-inflammatory genes in the spinal cords of mSOD1 and *Cx30*KO-mSOD1 mice. (**B**) Cluster analysis of gene expression arrays on pan-reactive, A1-specific, and A2-specific genes in the spinal cords of mSOD1 and *Cx30*KO-mSOD1 mice.

## Data Availability

Not applicable.

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
