# Peer review of "Connexin 30 Deficiency Ameliorates Disease Progression at the Early Phase in a Mouse Model of Amyotrophic Lateral Sclerosis by Suppressing Glial Inflammation"

_ijms, 2022, doi:10.3390/ijms232416046_

Round 1
Reviewer 2 Report
In the present study, the authors investigated the role of Cx30 in the mSOD1 mice.
This study is interesting, unfortunately, this manuscript needs improvements and corrections before publishing may be possible.
General points:
Please add a list of abbreviations before References section to your manuscript.
Please add as a Figure 1 a timeline of all your experiments.
Please add a Conclusion and the Future perspectives sections to your manuscript.
Special points:
Keywords: please add also to keywords: mSOD1 mice; amyotrophic lateral sclerosis
Introduction
Lines 30-41: please add multiple references at the end of each these sentences.
Lines 46-63: please describe all these sentences exactly.
Line 57: please add multiple references at the end of this sentence.
Lines 64-68: please add multiple references at the end of each these sentences and please describe all these sentences exactly.
Please add your hypotheses of this study at the end of your Introduction section.
Results
Please describe all behavioural tests used in your study very exactly separately in your Materials and Methods section.
Discussion
Important: please describe all studies from the literature referred by you more exactly and compare these studies more exactly with your results.
Lines 256-266: please describe all these sentences exactly.
Lines 276-282: please describe all these sentences exactly.
Materials and Methods
Lines 313-316: please add references at the end of this section.
Please add also the exactly date for the permission of all your experiments.
Line 319: please add an exactly total animals number used in your study. Did you use a female or male mice? How much of each gender?
Please add an exactly number of the animals use in each section.
Please add, according to which group or publication you did each method.
Round 2
Reviewer 1 Report
There are some mistakes in the new version of the manuscript:
-pages 6, 9, 12, 15 and 18 present old figures. Please remove them.
- in figure 2F, why statistical analysis is different in the anterior and dorsal horn? differences in anterior horn between mSOD1 and Cx30KO mSOD1 are really small to declare that there are some differences.
-statistical analysis is not well indicated in figures 3C and 3D and 3G, 3I and 3J. Please, authors should check it.
- Figure 5 have 3 new graphs and there are no letter for them. Indicate statistical differences for 20 W in the first graph (for S100A10). Check also statistical analysis for graph GFAP/actin, specially for 20W condition.
-Indicate statistical analysis in graph of figure 6E (iba1/actin), specially for 20w condition.
Reviewer 2 Report
Thank you for your corrections.
This manuscript needs once more some improvements and corrections before publishing may be possible.
Special points:
Introduction
Lines 31-32: please add multiple references at the end of this sentence.
Materials and Methods
Behavioural testing:
Please describe each behavioural test used by you as a separate section and very exactly. Please add to each these section the appropriate references according to which publication or group you did this test.
